# Synthesis, In Silico, and Biological Evaluation of a Borinic Tryptophan-Derivative That Induces Melatonin-like Amelioration of Cognitive Deficit in Male Rat

**DOI:** 10.3390/ijms23063229

**Published:** 2022-03-17

**Authors:** Mónica Barrón-González, Martha C. Rosales-Hernández, Antonio Abad-García, Ana L. Ocampo-Néstor, José M. Santiago-Quintana, Teresa Pérez-Capistran, José G. Trujillo-Ferrara, Itzia I. Padilla-Martínez, Eunice D. Farfán-García, Marvin A. Soriano-Ursúa

**Affiliations:** 1Academias de Fisiología, Bioquímica Médica, Sección de Estudios de Posgrado e Investigación, Escuela Superior de Medicina, Instituto Politécnico Nacional, Plan de San Luis y Diaz Mirón s/n, Col. Casco de Santo Tomás, Alc. Miguel Hidalgo, Mexico City 11340, Mexico; monicabarronglz@gmail.com (M.B.-G.); abadantonio19315@gmail.com (A.A.-G.); martinsnqn3@gmail.com (J.M.S.-Q.); tperezc@ipn.mx (T.P.-C.); jtrujillo@ipn.mx (J.G.T.-F.); 2Laboratorio de Biofísica y Biocatálisis, Sección de Estudios de Posgrado e Investigación, Escuela Superior de Medicina, Instituto Politécnico Nacional, Plan de San Luis y Diaz Mirón s/n, Col. Casco de Santo Tomás, Alc. Miguel Hidalgo, Mexico City 11340, Mexico; mrosalesh@ipn.mx; 3Departamento de Nefrología, Hospital General de México, “Dr. Eduardo Liceaga”, Dr. Balmis 148, Alc. Cuauhtémoc, Mexico City 06720, Mexico; ocamponestoral@hotmail.com; 4Laboratorio de Química Supramolecular y Nanociencias, Unidad Profesional Interdisciplinaria de Biotecnología del Instituto Politécnico Nacional, Av. Acueducto s/n Barrio la Laguna, Ticomán, Mexico City 07340, Mexico; ipadillamar@ipn.mx

**Keywords:** boron, Alzheimer, boroxazolidone, melatonin, male rat, androgen depletion, cognitive deficit, boron-containing compounds, amyloid, melatonin receptors

## Abstract

Preclinical and clinical evidence supports melatonin and its analogues as potential treatment for diseases involving cognitive deficit such as Alzheimer’s disease. In this work, we evaluated by in silico studies a set of boron-containing melatonin analogues on MT1 and MT2 receptors. Then, we synthesized a compound (borolatonin) identified as potent agonist. After chemical characterization, its evaluation in a rat model with cognitive deficit showed that it induced ameliorative effects such as those induced by equimolar administration of melatonin in behavioral tests and in neuronal immunohistochemistry assays. Our results suggest the observed effects are by means of action on the melatonin system. Further studies are required to clarify the mechanism(s) of action, as the beneficial effects on disturbed memory by gonadectomy in male rats are attractive.

## 1. Introduction

Alzheimer’s disease (AD) and other dementias with a high global burden involve cognitive deficit [1]. The disruption of the cholinergic system, amyloid-beta (Aβ) accumulation, and the formation of tangles are core in the current approach to explain the etiology of AD and for generating AD treatment [2,3]. However, there are neurotransmitters systems that have been recently involved in the genesis of the disease, among these are those using monoamines as potential therapeutic agents [4].

Melatonin and its receptors (MT1 and MT2, belonging as they do to G-protein coupled receptors, GPCR) have attracted attention. This is due to some reports suggesting a key role for this monoamine. Among these are those reporting that melatonin might be an interesting biomarker, showing an inverse correlation between the levels of melatonin in cerebrospinal fluid and the severity of the neuropathology [5]. However, there also are authors who report melatonin as a potential therapeutic agent because it can ameliorate the formation of Aβ-plaques and neurofibrillary tangles.

Moreover, melatonin administration improves the behavioral impairments related to AD in murine models, including disrupted circadian rhythm, cognition, learning, memory, motor function, mood, sleep, and stress response [6,7]. Furthermore, melatonin induces beneficial effects on the cholinergic system by increasing acetylcholine release and inhibiting choline acetyltransferase [7]. Furthermore, it also appears to exert benefits linked to the modulation of other monoamine systems, such as the serotonergic and dopaminergic systems by means of unestablished mechanisms [7,8].

Additionally, the chemical structure of melatonin is related to structures acting as antioxidants or regulators of inflammation [9,10]; in fact, its administration limits the excessive influx of Ca^2+^ and the excessive efflux of Mg^2+^ related to some inflammatory and neurotoxicity mechanisms [7,8].

On the other hand, boron-containing compounds (BCCs) possess expanding actions as therapeutic agents [11,12,13]. Today, boric acid, bortezomib, tavaborole, crisaborole, and vaborbactam are allowed to use in humans, but many other BCCs are showing attractive effects as potential treatments in human diseases, including neurological diseases. The mechanisms for potential therapeutic action of BCCs in the nervous system involve neuronal action through enzyme inhibition [14], acting as ligands of membrane receptors [13,15], regulating immune response [16], glial activities [17], and levels and the action of brain-derived growth factors [18,19]. Thus, action of BCCs on neuronal disease models are increasing [20,21]. Among these are those suggesting multitarget beneficial action [22], those showing a pro-survival effect on human-derived SH-SY5Y cells such as an Aβ-toxicity model [15,23], as well as the ameliorative properties in in vitro and in vivo models of AD [24,25].

In this work, the synthesis of a borinic compound (borolatonin) structurally related to melatonin (Figure 1) is reported. In addition, its theoretical ability to act as a melatonin-receptor ligand, and its effects in acting on the neuronal changes and cognitive deficit induced by gonadectomy in rats were evaluated. Moreover, the effects were compared to those induced by melatonin administration at an equimolar dose.

## 2. Results and Discussion

### 2.1. Chemistry

The synthesis proceeded as expected, in agreement with a similar process reported by Ocampo-Néstor for a structurally related dopa-derivative (Figure 2) [26].

Synthesized (4-((1*H*-indol-3-yl)methyl)-5-oxo-2,2-diphenyl-1,3,2-oxazaborolidin-2-uide (Borolatonin) is a pearl color solid, soluble in ethanol, DMSO and 1 mg/mL ethanol:water 1:9 or DMSO:water 7:3. (65.1% yield); m.p: 260–262 °C; IR νmax (cm^−1^): 424, 706, 748, 971, 1432, 1596, 1710, 3411. Raman shift (cm^−1^): 760.9, 1000.2, 1022.6, 1152.4, 1339.8, 1552.4, 1602.6, 1726.37, 3227, 3285.8, 3535. ^1^H NMR (300 MHz; D6-DMSO) δ 9.23 (1H,s, H7-indole), 7.33 (4H, ddt, J = 8.1, 6.3, 1.5, H10-13), 7.26–6.93, 10H, m, H17-21,H17′-21′), 6.67–6.51 (3H, m, H4, H3-ammonium), 3.52 (2H, J = 8.5 Hz, H7-indole), 2.98 (1H, dd, J = 14.4, 3.9 Hz, H6), 2.73 (1H, dd, J = 14.5, 10.3 Hz, H6′). ^13^C NMR (75 MHz; D6-DMSO) δ 174.01 (COOR, C5), 156.61 (C16, ipso), 131.57 (C14), 131.23 (CH, o), 130.66 (CH, C15), 127.52 (CH, m), 127.44 (CH, p), 127.37 (CH, C12), 126.37(CH, C11), 126.28(CH, C10), 115.68(CH, C9), 57.47 (CH, C4), 34.59 (CH2, C6). ^11^B NMR (128 MHz; MeOD) δ 5.26. MS *m/z* 391.77 [M + Na]+. Elem.Anal. calcd. For C_23_H_21_BN_2_O_2_: C 75.02%, H 5.75%, B 2.94%, N 7.61%, O 8.69% found: C 74.29%, H 5.68%, N 7.50%.

Chemical characterization allows us to identify the addition of the boron moiety to the amino acid segment of L-tryptophan, and we can also identify the mode of the reaction at the amino terminus. The signal at δ 5.26 ppm in the ^11^B spectrum supports the tetracoordination of boron in the structure of boroxazolidone.

### 2.2. Molecular Modeling

The models for rat MT1 and MT2 melatonin receptors were built from those crystallized for the human homologues (PDB CODES: 6ME5 for MT1-agomelatine, and 6ME9 for MT2-ramelteon complexes). Building was successful judged by the structural assessment; in agreement, Ramachandran plots (Appendix A) showed more than 96% of the residues in allowed areas. Only a few differences were found between the orthosteric binding region of human and rat receptors (Figure 3).

#### 2.2.1. Ligand Interactions with Melatonin Receptors

The re-docking of the removed agomelatine or ramelteon yielded similar docked poses from that in the crystal structure (RMSD < 0.697 Å between docked and the respective crystallized ligand, Figure 4) in the modeled receptor, supporting the ability of our method to estimate the adequate binding position. Similar poses were found for the tested ligands in any of the crystallized or built models. In fact, the ligands with known activity were able to reach the orthosteric site and the adjacent regions, as has been described previously [27,28,29,30].

In the highest affinity complexes, melatonin, agomelatine, ramelteon, and borolatonin were able to reach the orthosteric site or a nearby region; however, alternative docking poses (with minimal difference in energy score) showed all of these as inside the orthosteric site in a similar pose to that observed in crystal complexes. In fact, for borolatonin (Figure 5), simulations on any of the receptors revealed interactions at the orthosteric site. The most common interactions are through Van der Waals and hydrophobic types for any of these compounds. For borolatonin, interactions with three or four aromatic rings from the residues in the binding site comprised a common datum. In addition, the hydrogen bonds between the amine in the indole moiety of the compounds and a threonine or tyrosine residue were shared in the complexes; the exposed oxygen from the boron-containing ring is also able to interact. Conversely to that reported for other boron-containing compounds in complexes with metabotropic receptors, including amine receptors [21,26,31], the boron atom (marked with an asterisk in Figure 5) exhibited no direct interaction with any residue in the binding sites.

Regarding residues that were different among homologue receptors, only the tryptophan residue in position 254 for MT1 and 264 for MT2 receptors appears to be able for add interactions in the binding pockets; specifically, they permit the majority of the ligands to form pi–pi interactions with sidechain residues in the rat models. However, the binding pocket volume is similar in all four models.

#### 2.2.2. Affinity Estimation

The estimated affinity by docking procedure was related with previous experimental measurement of human receptors (Figure 6, experimental values were taken from the IUPHAR-database [32]). The coefficient of determination for 21 known ligands on MT1 receptor and 19 known ligands on the MT2 receptor (Appendix A), was 0.8861 for values on MT1, while this was 0.7554 for values on MT2 (only the 5-HEAT ligand was identified as an outlier on both receptors). All affinity values were underestimated by 3–4 orders of magnitude. This is in agreement with similar procedures for the affinity estimation in G-protein-coupled receptors [21,29,33].

Albeit no direct interactions of the boron atom were observed with residues in the receptors, the majority of the tested BCCs demonstrated higher estimated affinity than the known ligands of melatonin receptors (Figure 7). In particular, higher affinity was identified for BCCs with three aromatic rings in their structure, in addition to the five-member ring characteristic of the boroxazolidones, among these borolatonin.

### 2.3. Behavioral Changes by Hormone Deprivation and the Effects of Melatonin or Borolatonin

The diminished cognitive performance in passive avoidance task (PAT) was observed after three weeks of orchiectomy as reported elsewhere [34]. Thus, a significant decrease in latency into the safe compartment was registered in comparison with homogeneous pre-surgical gonadectomy behavior. These shorter latencies are related to the decreased levels of androgens in rats [35] and are in line with memory disruption related to androgens decreasing in humans [36,37]. In rats, steroid decreasing causes effects in the synthesis and degradation of key neurotransmitters such as acetylcholine, catecholamines, and other monoamines [8,38,39], including melatonin [38,39].

The administration of the melatonin or borolatonin compound increased retention latency in the evaluation of short-term (Figure 8A, *p* < 0.05) and the long-term (Figure 8B, *p* < 0.05) memory. These results are attractive, and are in line with the effect of melatonin administration on rats with induced memory impairment [40], including memory disruptions linked with steroid suppression by gonadectomy [41,42]. In fact, the common effect comprises limiting short-term memory impairment [43,44]. Moreover, effects on the long-term evaluation (as in the present work) have also been reported [45]. It is noteworthy that the similarity between the improvement induced by melatonin and by borolatonin, i.e., borolatonin, exerted a melatonin-like improvement judged by the behavior performance. In this regard, the effects of other compounds with analogous structure have also been reported in the early development of such compounds, such as agomelatine [46] or piromelatine (also denominated Neu-p11) [47]; at present, both compounds are known as ligands of melatonin receptors, acting as agonists with nanomolar affinity [47,48]. The mechanisms of action for melatonin and melatonin receptors involved in the cognitive deficit are unclear, but several neurotransmitters and hormones are disrupted by castration (including melatonin) [49], and there is no doubt that melatonin plays a key role modulating several memory processes and the production of neurotransmitters and receptors expression [50].

In motor performance in the open-field test in the pre-surgical, pre-treatment, or post-treatment evaluations no significant differences were observed (Figure 9, all *p* > 0.13). This motor evaluation is key to detect or discard motor disruption or anxiety behavior affecting memory performance.

### 2.4. Effects on Neuronal Survival and Beta Amyloid Presence in Hippocampus from Immunohistochemistry Assays

There were significant differences between the control and gonadectomized groups in all measured immunohistochemistry assays (Figure 10 and Figure 11, F ≥ 11.28, *p* < 0.01). In fact, the gonadectomy and the consequent hormonal deprivation reduced neuronal survival and increased the tissue changes associated with amyloid presence, as has been reported [51,52,53].

Regarding NeuN-positive cells, the administration of melatonin or borolatonin did not reduce the loss of neurons by androgen deprivation (F > 35.23, *p* < 0.01 vs. the control group). However, it should be considered that treatment was administered at three weeks post-orchiectomy; meaning that the loss of neurons had occurred as reported [54]. Additional studies are required to evaluate whether early administration of melatonin or borolatonin treatment could limit neuronal loss.

In fact, our results are in agreement with those showing that the administration of melatonin-receptors agonists, such as agomelatine [46], ramelteon [55], or Neu-P11, limited neuronal loss and the changes suggesting the presence of Aβ [47]. In the analysis of both Aβ presence and cells marked with NeuN, our results revealed that the administration of melatonin and borolatonin exert neuroprotection in hippocampal areas. In point of fact, melatonin or borolatonin induced a limit to any changes suggesting the presence of Aβ, with no difference in some areas of the hippocampus; that is, the treatments limited the Aβ in the entire hippocampus. Regarding correlation, considering all of collected data, the Spearman rho value was −0.879, suggesting an inverse relationship between the presence of Aβ and the total number of cells marked with NeuN in the analyzed hippocampal areas of the gonadectomized groups.

Immunohistochemistry and behavioral results are congruent, as these groups with improved memory performance exhibited decreased changes related to the presence of amyloid; even though poor changes appear in neuronal survival. Additional brain areas should be explored for the relation to this cognitive improvement.

Although the mechanisms involved in the observed behavior and neuronal changes induced by borolatonin administration in this cognitive deficit by hormonal deprivation must be clearly established, the observed changes are interesting in this first approach. Moreover, borolatonin is an attractive compound, considering that BCCs have shown increased affinity on the receptors of their analogous boron-free compounds [31], which could be in line with the findings in the current work, where the effects were similar to the number of borolatonin molecules, which was lower than that of the analogous melatonin (albeit they were 10 mg/kg each, due to the molecular weight of borolatonin being higher for borolatonin than for melatonin), but also, due to the increased half-time and prolonged action proposed for BCCs [13]. Diverse limitations exist, confirming the role of direct interaction on melatonin receptors, even if theoretical evaluation strongly suggest the ability of borolatonin to act on them. Additional neurotransmitter systems could be involved, among these, the serotoninergic, neurokinin, and histamine receptors, as well as certain structural features of borolatonin, are shared with known ligands and an evaluation of Swiss-target prediction servers suggest it (Appendix A). In addition, the use of specific antagonists of each system and binding assays on cells or tissues with high expression are desirable for supporting or discarding the main role of melatonin receptors [48].

Future research may be directed towards complementary evaluation of the effects of borolatonin actions on naïve animals and those with others mechanisms to induce cognitive deficit. Hence, the role of borolatonin as a melatonin receptor agonist and its selectivity should be evaluated by means of binding assays; as well as by the use of antagonists of the melatonin receptors, such as luzindole, which should be tested in terms of the observed effects on the steroid deprivation. Additionally, it would be interesting to evaluate the effects of this new compound when the memory impairment is mainly due to estrogen deprivation (i.e., in ovariectomized female rats). Additionally, the exploration of alternative mechanisms of action should be evaluated as compounds with similar chemical structure that have action on serotonin receptors [47], on ionic channels [56], or in the expression of brain-derived neurotrophic factor [57]. Moreover, if the action by interaction on the melatonin receptors is confirmed, evaluation of transduction pathways should be carried out, as some indole ligands have exhibited effects on classic G-protein-coupled pathways, but also on β-arrestin or calcium-dependent pathways [27].

## 3. Materials and Methods

### 3.1. In Silico Assays

#### 3.1.1. Molecular Modeling

To estimate the affinity and binding poses of both well-known ligands of melatonin receptors and the boron-derivatives on melatonin human and rat receptors, 3-dimensional structures were used. The models for human MT1 and MT2 were retrieved from the protein data bank (PDB, CODES: 6ME5 for MT1-agomelatine, and 6ME9 for MT2-ramelteon)) [58,59]; while the human homologues were built from those crystallized. Ligands, and co-crystallized water molecules were removed from the structures and the protein hydrogen atoms were added before submitting these receptors to docking analysis as mentioned elsewhere [60].

#### 3.1.2. Ligand Retrieval

Well-known ligands of melatonin receptors, as well BCC (essentially boroxazoli-done compounds with structural relationship to alpha-amino acids), were docked on the MT1 and MT2 receptors of human or rat. For each ligand the 3-D structure was built and geometrically optimized at the B3LYP/6-31G* level with Gaussian 09 soft-ware, Gaussian Inc., Wallingford, CT, USA [61].

#### 3.1.3. Docking Methodology

To identify the MT-R recognition sites and determine ligand affinities, docking simulations were performed. For that, all rigid/flexible bonds, partial atomic charges (Gasteiger–Marsili formalism), and non-merge hydrogens of the ligands were assigned. The Kollman partial charges for all atoms in the receptors, and the non-merge hydrogens were added using AutoDock Tools 1.5.6, maintaining the rest of the other program’s default parameters. Docking simulations were performed utilizing a commonly-used search algorithm (hybrid Lamarckian Genetic) implemented on AutoDock version 4.2.6 software, Scripps Research, San Diego, CA, USA [60]. The initial population was of 100 randomly-placed individuals, and the maximum number of energy evaluations was 10 million. To search for all potential binding sites on the MT-R, input initializations of the ligand structures and MT-R binding site definitions were carried out using a GRID-based procedure [62]. A blind docking procedure using a box of 70 × 70 × 70 Å point grid with 0.375-Å spacing was utilized, centered in the middle of the alpha carbon of the residues at positions **^3.32^** and **^6.52^** considering Ballesteros–Weinstein numbering [63]. Docked orientations within a root-mean square deviation of 0.5 Å were clustered together and the lowest free-energy cluster returned for each compound was used for further analysis using AutoDock Tools version 1.5.6 software. The resulting complexes were visualized using VMD version 1.9.3 software, University of Illinois at Urbana-Champaign, Champaign, IL, USA [64]. The free energy and Ki values yielded from the Autodock program were considered for estimating affinity values as elsewhere [34].

### 3.2. Chemistry

#### 3.2.1. Chemicals

Melatonin (B6768), L-Tryptophan (T0254), 2-Aminoethyl diphenyl borate (D9754), and solvents (ethanol, ethyl ether) used in synthesis procedure were purchased from Sigma-Aldrich Co. (St. Louis, MO, USA).

#### 3.2.2. Synthesis of Borolatonin

In brief, L-tryptophan (1.0 g, 4.896 mmol) was dissolved in a minimal quantity of distilled water (Solution A), while 2-aminoethyl diphenylborinate (1.5 g, 6.664 mmol) was dissolved in 25 mL of ethanol; after intense and constant stirring, the concentrated HCl (37%, 2 mL) was added to release the diphenylborinic acid, and distilled water was then added until a milky suspension appeared (Solution B). The diphenylborinic acid was immediately and quickly extracted using ethyl ether that was mixed with Solution A. The extraction of diphenylborinic acid was repeated until the milky suspension of Solution B disappeared. Ethanol was added to the mixture of solution A and diphenylborinic acid until the formation of a homogeneous mixture, the mixture was left to reflux for 3 h, and the solvent was extracted utilizing a Dean–Stark trap at 70 °C.

The crude reaction was added to distilled water to precipitate, after 24 h at room temperature, the pearl-colored product was washed alternately with distilled water and a 7:3 hexane–acetate mixture, filtering the product under vacuum and drying.

### 3.3. Behavioral Evaluation

To evaluate the potential of BCCs for exerting ameliorative therapeutic effects in AD, experimental assessment was conducted in an murine model of AD, with cognitive deficit induced by hormonal diminishing after gonadectomy [54,65]. The evaluation was in agreement with Mexican National Research Council specifications for the production, care, and use of laboratory animals and the use of some species inside the laboratory; the protocol was approved by the Institutional Committee for the Care of Animals in the Laboratory (CICUAL-01/23-01-2019-1.0).

Thirty-two male Wistar rats (10-week-old and weighing 200–220 g) were obtained from the Bioterium of Centro de Investigacion y de Estudios Avanzados del Instituto Politécnico Nacional. Rats were maintained under standard conditions with access to food and water ad libitum. They were free of known viral, bacterial, and parasitic pathogens, and were housed in polystyrene cages in a room under controlled humidity (50 ± 5%), temperature (20–25 °C), and lighting (12 h light/dark cycle, lights on at 7 am). For the surgical procedure, rats were anesthetized with pentobarbital (40 mg/kg/intraperitoneally (i.p.)).

The rats were randomly distributed into four groups with eight animals in each one. Three groups were gonadectomized; orchiectomy was performed through ventral incision in the scrotum, as described elsewhere. Treatment was administered at 21 days post-gonadectomy. Control (sham) and gonadectomized control groups were treated with vehicle (saline solution 0.9%, 10 mL/kg of body weight (b.w.)) daily for 28 days. On the same days, the remaining two groups were treated with melatonin (i.p., 10 mg/kg, 4.11 × 10^−5^ mol) or borolatonin (i.p., 10 mg/kg, 2.71 × 10^−5^ mol) in the same volume of vehicle [66,67]. All of the behavioral evaluations were carried out between 10:00 and 14:00 h.

The open field test was conducted for 5 min prior to treatments and memory evaluation to discard any motor disturbance affecting performance. In brief, rats were placed into motor activity measuring cages (50 cm × 50 cm × 50 cm, with detectors each 2.5 cm, OA-BioMed OMNIALVA^®^, Mexico City, Mexico) for determining total number of movements, distance traveled, highest speed, and vertical movements as elsewhere [34]. Treatment were in a similar posology to those reported as active in rat memory evaluation [66,67].

For evaluating memory, the passive avoidance test (PAT) was performed for each group as reported previously [34,54]. In brief, we used an apparatus (OMNIALVA^®^, Mexico City, México) with two compartments, including the safe (white) and the shock (black) compartments. Each animal was trained in the safety compartment for 10 s, and then the sliding door was lifted. Latency for crossing the threshold to the shock compartment was recorded (acquisition latency), excluding animals that spent more than 100 s to cross to the other side. Once the animal crossed into the next compartment, the door was closed and a 3 mA foot-shock was delivered for 5 s. The door was opened, and the time that the animal took to return to the safe compartment was measured, this was considered escape latency. For memory evaluation, the animal remained there for 30 s prior to being returned to its individual cage. Either 10 min (considered as short-term memory) or 24 h (considered as long-term memory) after this training procedure, a retention test was performed. The animal was again placed in the safety compartment for 10 s, and after the door was opened, the time that the animal remained in the safety compartment before entering the shock compartment was recorded (retention latency). The test was considered over when the rat either entered the shock compartment or remained in the safety compartment for 600 s. Memory evaluation was performed one day before the surgical procedure, and after the treatment (three weeks after surgical procedure).

### 3.4. Immunohistochemistry Assays

After the behavioral evaluation, rats were anesthetized with pentobarbital and perfused transcardially with 200 mL of ice-cold (4 °C) sodium phosphate buffer (PBS) 0.1 M, pH 7.4, and a freshly prepared fixative consisting of 4% paraformaldehyde in 0.1 M phosphate buffer, pH 7.4. The brain was removed immediately after perfusion and post-fixed in ice-cold (4 °C) 4 % paraformaldehyde, followed by 48 h in ice-cold (4 °C) 30% sucrose in 0.1 M phosphate buffer for cryoprotection. Coronal slices of brain (30 μm thick) were cut on the freezing microtome and stored at −20 °C in a solution containing 30% ethylene glycol, 30% glycerol, and PBS until immunohistochemical analysis.

Free-floating sections were incubated in PBS containing 3% H_2_O_2_ for 10 min and rinsed three times with PBS for 5 min. Subsequently, sections were incubated in 0.1% Tween 20 in PBS (T-PBS) for 10 min, then incubated in 0.5% bovine serum albumin (BSA, GE Health Care, UK) in T-PBS for 2 h. Finally, sections were incubated with the primary antibody anti-βA1-24 (1:1000, ABBIOTEC, CA, USA) or anti-NeuN (1:1000, Millipore, MA, USA) at 4 °C for 48 h. The following day, sections were rinsed three times in T-PBS for 5 min and incubated with a biotinylated secondary antibody and a peroxidase-labeled streptavidin reagent (LSAB System-HRP Kit; DAKO, CA, USA) for 30 min each. Antibody labeling was visualized by exposure to 3–3′-diaminobenzidine (DAKO, CA, USA). Omission of primary or secondary antibodies as negative controls resulted in the absence of immunolabeling. Finally, the sections were rinsed twice in PBS and mounted in Entellan solution (Millipore, MA, USA). The number of βA1-24 or NeuN-positive neurons was counted through random sampling bilaterally in two sections per animal under a light microscope (magnification 40×) by two independent trained observers blinded to the treatment for avoiding biased appraisal. Cell counts from the right and left hemispheres of each of the two sections were averaged to provide a single value for each animal. From these data, the mean of these labeled neurons in Cornu Ammonis (CA)1, CA2, CA3, and DG (Dentate Gyrus) hippocampal areas (*n* = 4) was calculated for each experimental group.

### 3.5. Statistical Analysis

Data were expressed as the mean ± the standard error of the mean (S.E.M.) One-way ANOVA followed by the Tukey post hoc test was utilized in the case of parametric data. For the data found in behavioral evaluation (Figure 8 and Figure 9), the Kruskal–Wallis test was employed as the data are not normally distributed, and comparisons were performed by the Dunn’s post hoc multiple comparison test. Differences among data in immunohistochemistry assays (Figure 10 and Figure 11) were identified by One-way ANOVA followed by the Tukey post hoc test. The Spearman’s rho rank correlation test was utilized to compare NeuN neuron counts and Aβ plaque counts. All statistical analyses were performed using Sigma Plot version 12.0 software. Statistical significance was set at *p* < 0.05.

## 4. Conclusions

A new boron-containing molecule was synthesized, characterized, and evaluated in the cognitive performance of rats. Hence, in silico assays suggested a tryptophan boron-containing derivative (4-((1*H*-indol-3-yl)methyl)-5-oxo-2,2-diphenyl-1,3,2-oxazaborolidin-2-uide) as a potent agonist on both, MT1 and MT2 melatonin receptors. This is in line with the neuroprotective effects induced by intraperitoneal administration of this newly synthesized compound in behavioral tests and immunohistochemistry assays. In fact, the similar ameliorative effects observed by the administration of melatonin suggested the observed effects are by means of action on the melatonin system. Additional in vitro and in vivo evaluation is required to support or discard this mechanism, as well as to evaluate whether or not some other mechanisms are involved.

## Figures and Tables

**Figure 1 ijms-23-03229-f001:**
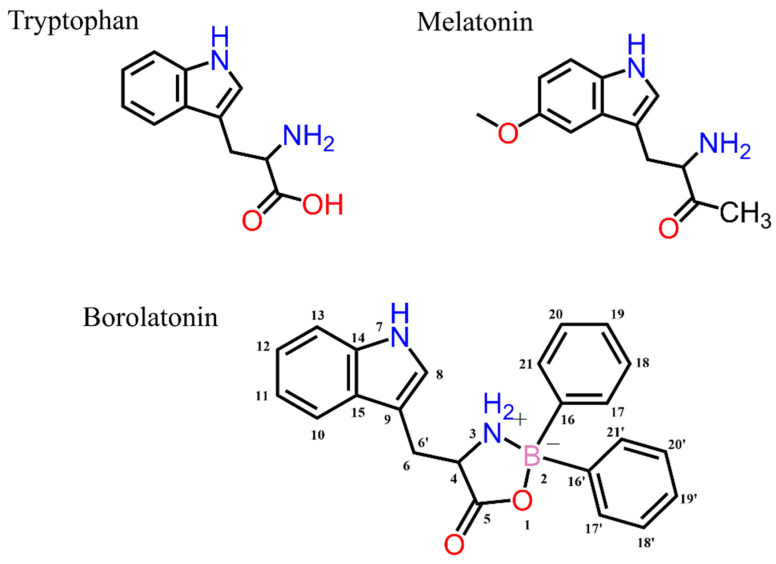
The chemical structures of tryptophan, melatonin, and borolatonin (adduct of tryptophan and 2-APB).

**Figure 2 ijms-23-03229-f002:**
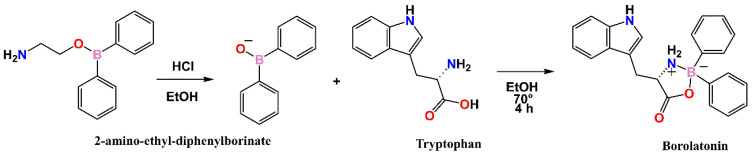
Synthesis procedure of the borolatonin compound.

**Figure 3 ijms-23-03229-f003:**
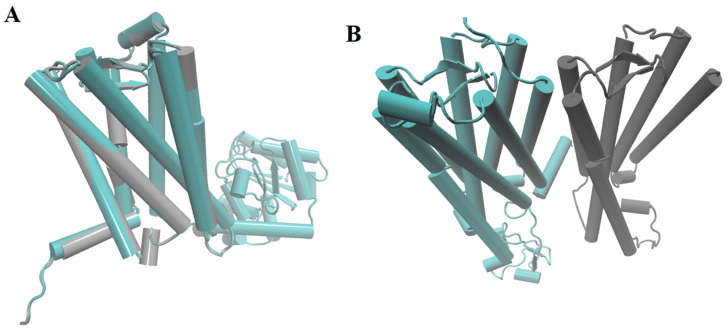
The 3-D models of human (cyan) or rat (gray) of MT1 (**A**) and MT2 (**B**) melatonin receptors. High similarity in the general disposition, conformations, and binding sites of the four receptors is noteworthy.

**Figure 4 ijms-23-03229-f004:**
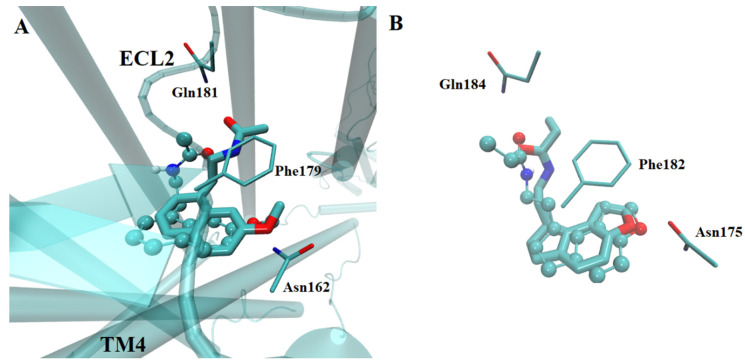
Overlay of docked ligand structures (in stick and balls representation) on human MT1 ((**A**), with agomelatine) and MT2 ((**B**), with ramelteon) melatonin receptors as they are found in the crystal complexes. The sidechain of some residues considered key in the binding pocket are depicted as reference. High similarity in the binding sites of both receptors is notable. Oxygen atoms are in red color, nitrogen atoms are in blue color.

**Figure 5 ijms-23-03229-f005:**
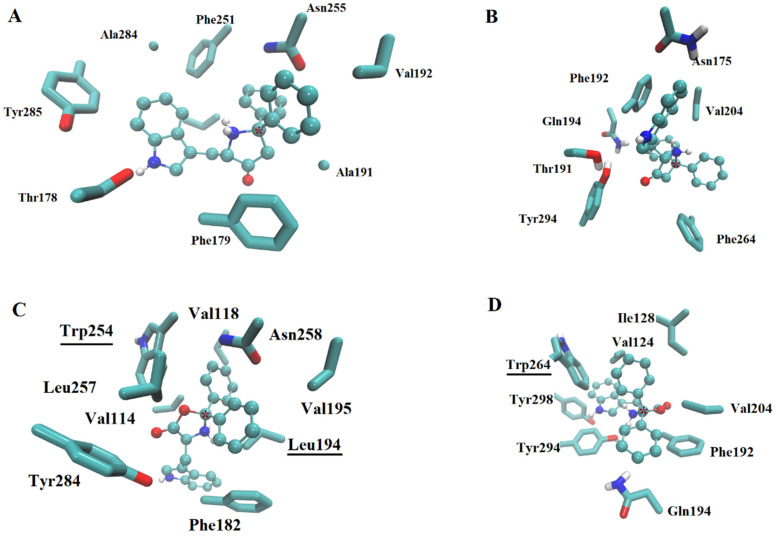
Docking of borolatonin (in sticks and balls representation) on melatonin receptors. Binding site on the crystalized human MT1 (**A**), MT2 (**B**), and the built models of rat MT1 (**C**) and MT2 (**D**). All sidechains of residues are labeled in licorice representation. Underlined residues marked in the lower panels are different in the binding site of the homologous human receptor. Asterisk is in the boron atom of each borolatonin molecule. In the D panel, Thr 191 and Asn 175 were not included for clarity. Oxygen atoms are in red color, nitrogen atoms are in blue color.

**Figure 6 ijms-23-03229-f006:**
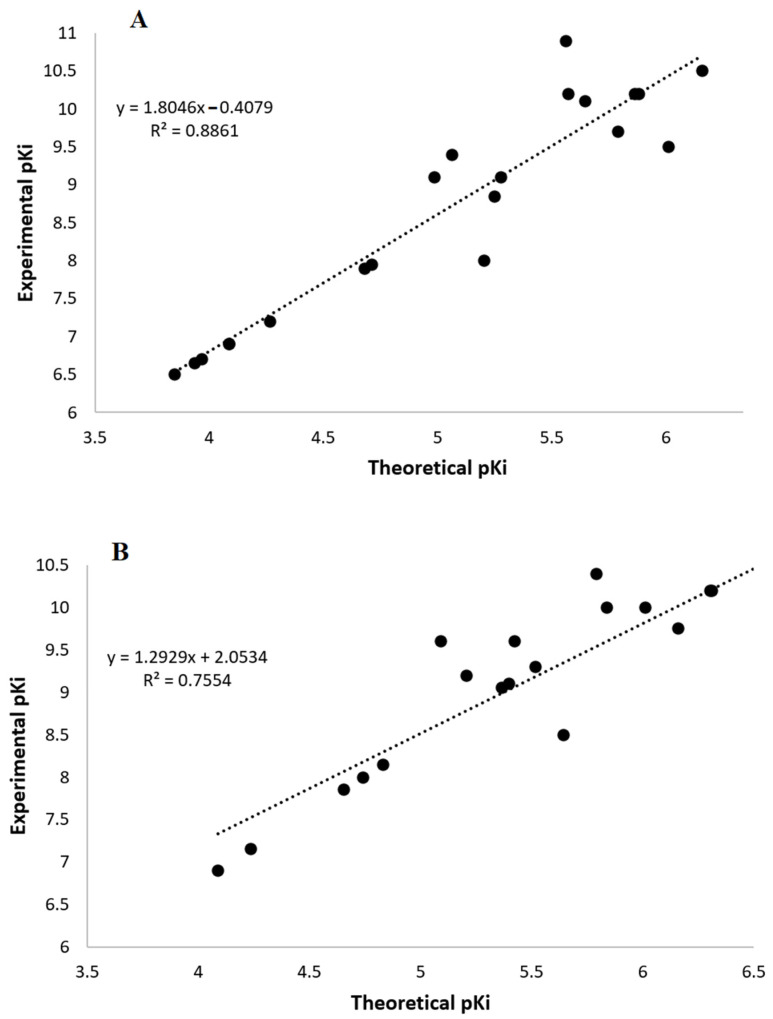
Relationship between estimated (theoretical) and reported (experimental) affinity of tested ligands on human MT1 (**A**) or MT2 (**B**) melatonin receptors. Linear regression equation and coefficients of determination are shown in each plot.

**Figure 7 ijms-23-03229-f007:**
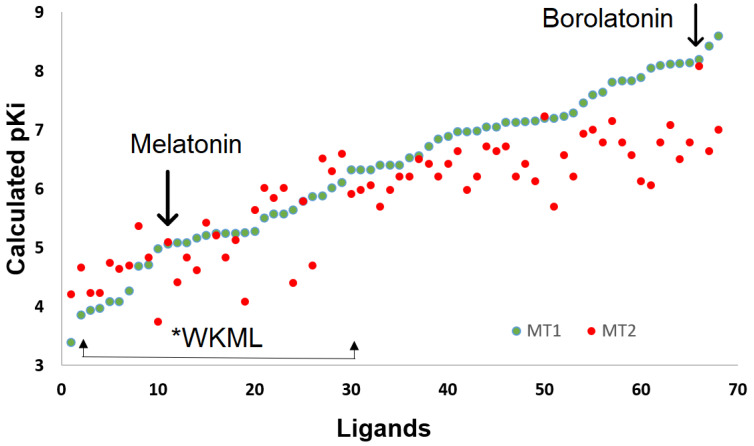
Affinity of tested ligands on human melatonin receptors. Arrows are for labeling the endogenous agonist melatonin, the tested compound borolatonin, and a region for predicted affinity values for the well-known melatonin-receptor ligands (*WKML).

**Figure 8 ijms-23-03229-f008:**
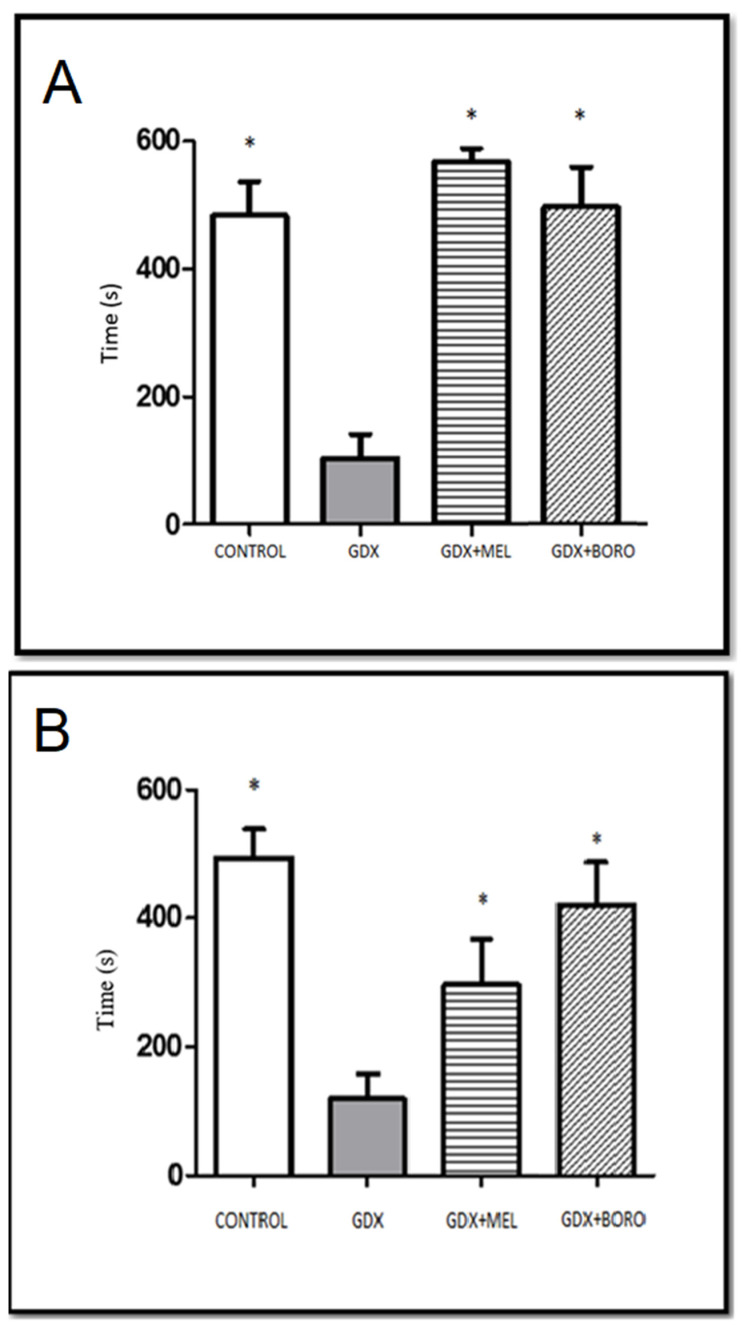
Post-treatment short-term (**A**) and long-term (**B**) memory performance. Results are expressed as mean of time in the safe compartment ± Standard Error of the Mean (SEM, *n* = 8). * *p* < 0.05, compared to orchiectomized control group (GDX); MEL, melatonin; BORO, Borolatonin.

**Figure 9 ijms-23-03229-f009:**
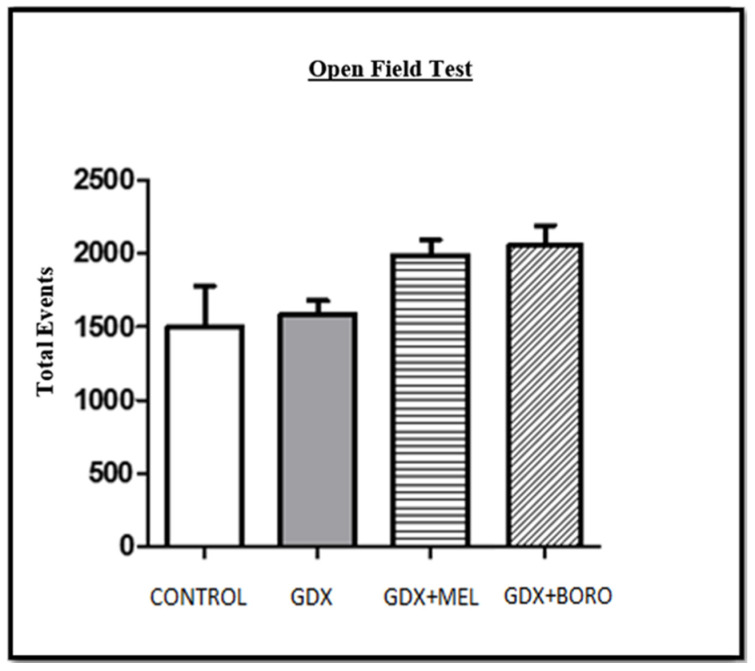
Motor performance in the open-field test at the post-treatment evaluation. No difference was found among groups. Results are expressed as mean ± SEM (*n* = 8). GDX, gonadectomized; MEL, melatonin; BORO, Borolatonin.

**Figure 10 ijms-23-03229-f010:**
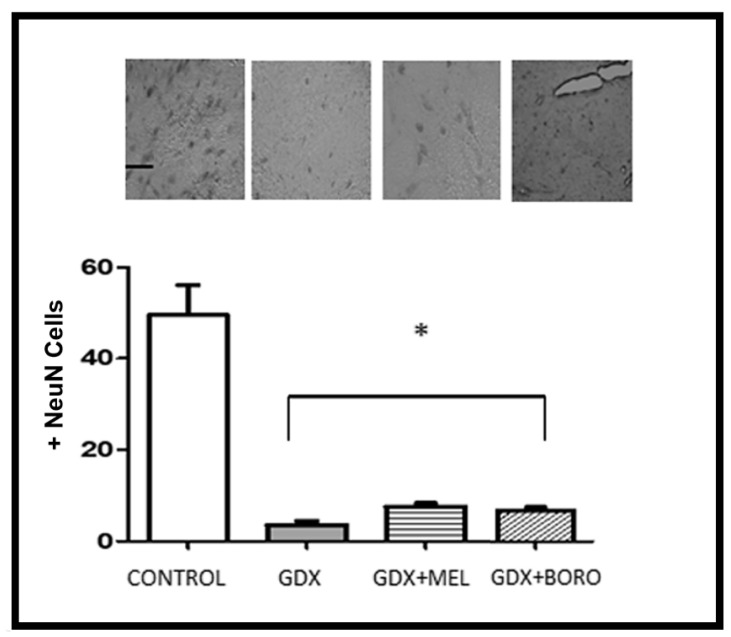
Neuronal survival. Above are photomicrographs of NeuN positive labeling observed in the hippocampus. Magnification 40×. Scale bar represents 100 μm. In the plot the columns represent the number of NeuN-positive neurons, bars represent SEM. Significant differences in groups compared with the control group (*n* = 4, * *p* < 0.01); GDX, gonadectomized; MEL, melatonin; BORO, Borolatonin.

**Figure 11 ijms-23-03229-f011:**
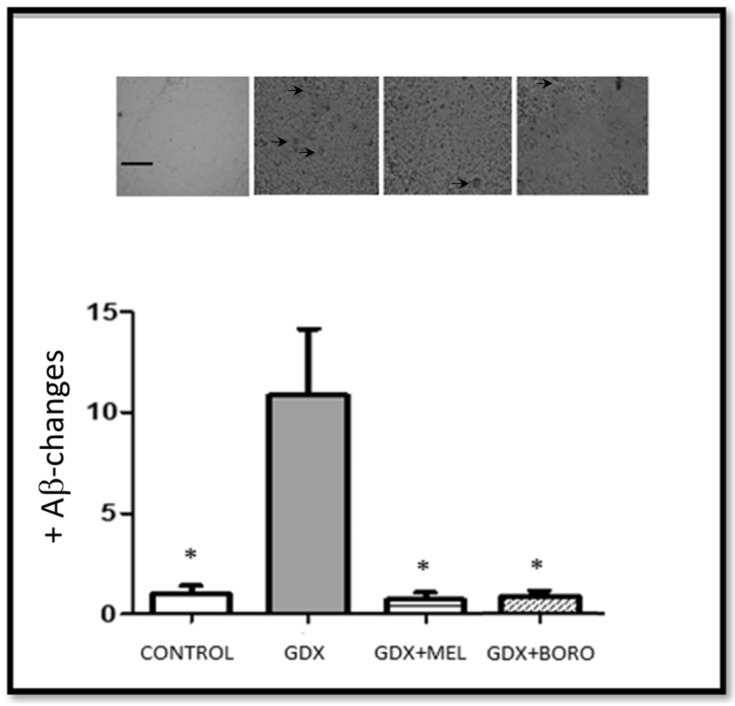
Amyloid beta accumulation. Above are photomicrographs displaying Aβ-positive labeling (arrows) seen in the rat hippocampus. Magnification 40×. Scale bar represents 100 µm. In the plot columns represent the number of positive Aβ-regions related to plaques in the hippocampus ± SEM (as bars). Significant differences in groups compared with to orchiectomized control group (GDX) group (*n* = 4, * *p* < 0.01); MEL, melatonin; BORO, borolatonin.

## Data Availability

Any complementary data of the reported data are available on request from the corresponding authors.

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
