# Peer review of "Synthesis, In Silico, and Biological Evaluation of a Borinic Tryptophan-Derivative That Induces Melatonin-like Amelioration of Cognitive Deficit in Male Rat"

_ijms, 2022, doi:10.3390/ijms23063229_

Round 1

Reviewer 1 Report

The manuscript entitled “Synthesis, in silico and biological evaluation of a borinic tryptophan-derivative which induces melatonin-like amelioration 3 of cognitive deficit in male rat.” is interesting research to find novel therapeutic agents for Alzheimer’s disease.

However, I have several comments.

It is hard for readers to understand the context. Furthermore, the sentences are sometimes mistaken. For example, in figure 8, they described p≤0.05. Did it mean at least one value is p = 0.05?

The manuscript requires a check for English by a native speaker and the authors should recheck the manuscript.

In, Figure 10 and 11, the authors showed the immunohistochemistry, however, they did not present the area in the hippocampus. CA1?, CA2?, CA3, or DG? The authors should present each area to readers.

In Figure 11, I could not find ** or ***. I could find only *. What is *? It is no information in the figure legend. Is this legend the copy of another manuscript?

In methods, it is insufficient information on rat models. How many total samples are used? How old were rats? There was no information in this manuscript. The authors should describe in detail.

In results, it is not easy to understand for readers. For example, at line 223 “[Fig. 9, all p>0.13]”. Why did they describe 0.13?

They describe “Regarding NeuN-positive cells, the administration of melatonin did not reduce the loss of neurons by androgen deprivation (p<0.05). However, it should be considered that treatment was administered after three weeks of orchiectomy; it means the loss of neurons had occurred 235 as is reported”, however, they did not describe the treatment for the rat model in detail. When and how long, how was the method of administration? How many concentrations was each agent?

They described “In the analysis of both Aβ presence and cells marked with NeuN, our results showed that the administration of melatonin and borolatonin exert neuroprotection in hippocampal areas. Regarding correlation, considering all collected data, the Spearman’s rho value of -0.879 was obtained suggesting an inverse relationship between Aβ presence and the total number of cells marked with NeuN.” However, in what group is this result? How was the p value?

The authors should clarify the difference between melatonin and borolatonin. If there was no difference in the effects, then was it better to use melatonin? The authors should emphasize the difference between melatonin and borolatonin in discussion and conclusions.

Reviewer 2 Report

The manuscript entitled „ Synthesis, in silico and biological evaluation of a borinic tryptophan-derivative which induces melatonin-like amelioration of cognitive deficit in male rat” contains chemical, behavioral and molecular studies of new compound, borolatonin. The Authors  describe the process of borolatonin synthesis, then model in silico the melatonin receptors MT1 and MT2 and the ligand-receptor interaction. Moreover, behavioral tests were performed: the open field test, in which the locomotor activity of the animals was measured, and the passive avoidance test, in which the cognitive disturbances were measured. Finally, immunochistochemical determinations of NeuN positive cells and amyloid beta accumulation in hippocampal neurons were performed.

 The Authors choose the methodology correctly and presented their results in a clear manner. Due to the multitude of research included in this manuscript, it is very extensive.

I have some  comments:

  1. All results are missing the "F" value, only the "p" values are given.
  2. The Authors investigated the effects of melatonin and borolatonin in gonadectomized animals, unfortunately there is no group with melatonin or borolatonin alone. This is a serious mistake because it is not known how these substances act in naive animals.
  3. The title of chapter 4.3 should rather be "Behavioral tests". This chapter lacks information on the animals used in the experiments: what species, strain, number animals in groups, what living conditions, etc.  There is also a lack of precise information on the doses of drugs and the method of their administration.
  4. Figure 11 contains photos that are of poor quality / resolution, therefore the differences in individual groups are not visible.

Please fill in all missing information.

Round 2

Reviewer 1 Report

In this revised manuscript, the authors have adequately addressed the reviewer’s comments. The manuscript is now significantly improved.